# Bio-Responsive Carriers for Controlled Delivery of Doxorubicin to Cancer Cells

**DOI:** 10.3390/pharmaceutics14040865

**Published:** 2022-04-15

**Authors:** Gheorghe Fundueanu, Marieta Constantin, Mihaela Turtoi, Sanda-Maria Bucatariu, Bogdan Cosman, Maria Anghelache, Geanina Voicu, Manuela Calin

**Affiliations:** 1Department of Natural Polymers, Bioactive and Biocompatible Materials, “Petru Poni” Institute of Macromolecular Chemistry of the Romanian Academy, 700487 Iasi, Romania; marieta@icmpp.ro (M.C.); sanda.bucatariu@icmpp.ro (S.-M.B.); cosman.bogdan@icmpp.ro (B.C.); 2“Medical and Pharmaceutical Bionanotechnologies” Laboratory, Institute of Cellular Biology and Pathology “Nicolae Simionescu” of the Romanian Academy, 050568 Bucharest, Romania; mihaela.carnuta@icbp.ro (M.T.); maria.anghelache@icbp.ro (M.A.); geanina.voicu@icbp.ro (G.V.)

**Keywords:** poly(*N*-isopropylacrylamide), *1*-vinylimidazole, stimuli-sensitive polymer, drug targeting, LCST

## Abstract

The cellular internalization of drug carriers occurs via different endocytic pathways that ultimately involve the endosomes and the lysosomes, organelles where the pH value drops to 6.0 and 5.0, respectively. We aimed to design and characterize pH/temperature-responsive carriers for the effective delivery of the anti-tumoral drug doxorubicin. To this purpose, poly(*N*-isopropylacrylamide-co-vinylimidazole) was synthesized as an attractive pH/temperature-sensitive copolymer. Microspheres made of this copolymer, loaded with doxorubicin (MS-DXR), disintegrate in monodisperse nanospheres (NS-DXR) under conditions similar to that found in the bloodstream (pH = 7.4, temperature of 36 °C) releasing a small amount of payload. However, in environments that simulate the endosomal and lysosomal conditions, nanospheres solubilize, releasing the entire amount of drug. We followed the NS-DXR internalization using two cancer cell lines, hepatic carcinoma HepG2 cells and lung adenocarcinoma A549 cells. The data showed that NS-DXR are internalized to a greater extent by HepG2 cells than A549 cells, and this correlated with increased cytotoxicity induced by NS-DXR in HepG2 cells compared with A549 cells. Moreover, NS-DXR particles do not cause hemolysis and erythrocytes aggregation. Administered *in vivo*, NS-DXR localized in the liver and kidneys of mice, and the loading of DXR into NS resulted in the reduced renal clearance of DXR. In conclusion, the newly developed poly(*N*-isopropylacrylamide-co-vinyl imidazole) particles are biocompatible and may be introduced as carriers for doxorubicin to hepatic tumors.

## 1. Introduction

Cancer, one of the most common diseases in modern times, is among the leading causes of death worldwide. Surgery and chemotherapy are the most used methods for cancer treatment, unfortunately the latter with many side effects due to non-selectivity.

Targeting drugs to a specific location using controlled delivery systems seems to have solved this problem to some extent [1,2,3]. An appealing strategy for targeted delivery of chemotherapy drugs to cancer cells is the use of hyaluronic acid (HA)-based nanocarriers since they assure an improved drug delivery to tumors with increased expression of the CD44 receptor, a receptor recognized by HA [4,5]. Other systems were designed to transport the drug intact through the bloodstream and release it to the desired location due to the action of external stimuli such as pH [6], temperature [7], or redox potential [8]. Among them, pH-sensitive controlled release systems are the most common because they exploit the pH variation in tumor tissues [9,10]. In general, the extracellular pH of the normal tissues is in the range of 7.2–7.5, while that of tumors is slightly acidic, being in the range of 6.4–7.0 [11]. As follows, it is challenging to design drug delivery systems able to change their physicochemical properties in response to these small pH changes. Moreover, recent reports stated that the intracellular pH of tumoral cells is neutral or even slightly alkaline as compared with that of normal cells [12]. Nevertheless, cancer cells behave similarly with normal cells in the internalization process of particles by endocytosis. In both cell types, the particle internalization involves passage from endosomes to lysosomes, where the pH decreases from 6.0–5.5 to 5.0, respectively [13]. These differences are significantly higher and allow the design of pH-sensitive microparticulate systems for drug targeting. In most cases, these drug carriers are made of pH-sensitive polymers that possess ionizable groups at pH values at the frontier between physiological and pathological conditions [14,15]. Generally, pH-sensitive polymers contain weakly acidic (–COOH) [16] or weakly basic (–NH_2_) [17] functional groups. These polymers, in linear or cross-linked form, modify their physico-chemical properties such as solubility [18], self-assembly capacity [19], degree of swelling [20], etc., in response to pH variations. Actually, changes in physico-chemical properties of polymers are a direct consequence of protonation/de-protonation of functional groups. 

*1*-Vinylimidazole (VI) is an excellent candidate for obtaining pH-sensitive polymers because it has weakly basic amino groups with a pKa of around 6.0, a value found in the cell compartments after particle internalization [21]. To maximize site-specific delivery, the development of a polymer with multi-stimuli-sensitive characteristics is imperatively required. The copolymerization of VI with a temperature-sensitive monomer, such as *N*-isopropylacrylamide (NIPAAm), gives rise to copolymers possessing double sensitivity; for these copolymers, the physico-chemical characteristics are modulated both by pH and temperature [22]. 

As it is well-known, poly(NIPAAm) in aqueous solution possesses a sharp phase transition (lower critical solution temperature, LCST) at a value of about 32 °C [23]. Below the LCST, the polymer is soluble, while above the LCST, it precipitates. However, for copolymers of NIPAAm with VI, the LCST is expected to depend on the amount and ionization degree of VI. The protonation/deprotonation of VI could modulate the solubility properties of the copolymer that will be further exploited in carrier synthesis for the transport and release of the drug at the desired place. Till now, few studies related to the biomedical application of poly(NIPAAm-co-VI) copolymer are reported. For instance, poly(NIPAAm-co-VI) particles were synthesized by freeze-drying the copolymer aqueous solution [24]. These pH- and temperature-sensitive particles protect the drug molecules inside the stomach, whereas it will be released through sudden squeezing of the carrier in the small intestine (and/or colon). Cross-linked poly(NIPAAm-co-VI) hydrogels with different co-monomer molar ratios were also synthesized by free radical polymerization in aqueous solutions using ammonium persulfate as initiator and TEMED as accelerator [25]. The release of Rhodamine 6G, used as a drug model, entrapped in these hydrogels, has been influenced by the pH and temperature of the release fluids. 

Doxorubicin (DXR) is a widely used chemotherapeutic agent, and DXR nano-formulations, namely PEGylated (Doxil^®^, Lipodox^®^) and un-PEGylated (Myocet) liposomes are currently used in clinics [26]. The encapsulation of DXR into nanoparticles leads to the reduction in its side effects (e.g., cardiotoxicity, nephrotoxicity, myelosuppression) by increasing the accumulation of DXR into tumor tissue through enhanced retention and permeability (EPR) effects and lesser amounts in healthy tissues. Aiming to further reduce the cardiotoxicity and overcome the DXR resistance, numerous other nanoparticle-based delivery systems of DXR were developed [27,28].

Herein, we designed and developed “intelligent” micro-vehicles for efficient transport and delivery of DXR to tumoral cells. Poly(NIPAAm-co-VI) microspheres loaded with DXR (MS-DXR) disintegrate in monodisperse nanoparticles (NS-DXR) in physiological fluids. The nanoparticles protect the entrapped DXR in conditions similar to that found in the bloodstream (pH = 7.4 and T = 36 °C), while in slightly acidic environments, simulating the different pH values in cell compartments, the drug is nearly completely released. Further, the cytotoxicity of NS-DXR and the efficacy of internalization of both NS-DXR and free DXR were evaluated, in two human tumor cell lines, hepatocarcinoma (HepG2) cells and adenocarcinoma alveolar basal epithelial (A549) cells. The hemolytic activity, the blood compatibility, and the biodistribution of NS-DXR compared to free DXR after intravenous administration in C57BL/6J mice were also evaluated. 

## 2. Materials and Methods

### 2.1. Materials

*N*-isopropylacrylamide (NIPAAm), purchased from Aldrich Chemical Corp. (Milwaukee, WI, USA), was recrystallized with hexane. *1*-Vinylimidazole (VI), *N*,*N*′-azobisisobutyronitrile (AIBN), 1,4-dioxane, and cyclohexane were supplied from Fluka AG (Buchs, Switzerland). Soybean lecithin was obtained from Iassyfarm S.A. (Iassy, Romania). Doxorubicin hydrochloride (DXR) was supplied from Sigma-Aldrich (St. Louis, MO, USA) 

Reagents and consumables used to perform biological assays were purchased from different sources as follows: Dulbecco’s modified Eagle’s medium (DMEM), paraformaldehyde (PFA), and Triton X-100, were purchased from Sigma-Aldrich (Merck KGaA, Darmstadt, Germany); 2,3-Bis-(2-Methoxy-4-Nitro-5-Sulfophenyl)-2H-Tetrazolium-5-Carboxanilide (XTT), phenazine methosulfate (PMS), 4′,6-diamidino-2-phenylindole (DAPI), fetal bovine serum, and antibiotics were from Thermo Fisher Scientific (Waltham, MA, USA); cell culture dishes were from TPP^®^ (Trasadingen, Switzerland), black 96-well micro test plates, F-bottom were from Ratiolab (Ratiolab GmbH, Dreieich, Germany); phalloidin-Fluorescein isothiocyanate (FITC) was from Tocris Bioscience (Bio-Techne Ltd., Minneapolis, MN, USA). 

### 2.2. Poly(NIPAAm-co-VI) Copolymer Preparation

Poly(NIPAAm-co-VI) copolymer was prepared by free radical copolymerization of the corresponding co-monomers. For example, 10 mmol of NIPAAm (1.13 g) and 1.5 mmol of VI (0.141 g) were dissolved in 10 mL 1,4-dioxane. Nitrogen, in the dried state, was purged through the solution for approximately 60 min. Before starting the polymerization, 0.15 mmol AIBN (0.025 g) was added, and the reaction lasted 20 h at 70 °C. The resulting copolymer was precipitated into a large volume of diethyl ether; then filtered and dried under vacuum. Finally, the copolymer was dissolved in distilled water, dialyzed for 7 days at room temperature (20 ± 2 °C), and recovered by lyophilization.

### 2.3. ^1^H-NMR Analysis of Poly(NIPAAm-co-VI)

^1^H-NMR analysis was used to confirm the chemical structure of the copolymer and to determine the co-monomer molar ratio in the copolymer. The experiments were carried out in deuterated water on a Varian Mercury Plus 400/Varian VXR 200 spectrometer that operates at a frequency of 400 MHz. The molar ratio of co-monomers in the poly(NIPAAm-co-VI) copolymer was calculated using Equations (1) and (2): 9x + 2y = A1(1)2y = A2(2)
where x and y are the relative molar fractions of NIPAAm and VI, respectively. 

Equation (1) represents the total area (A1) of the peaks between 0.5 and 2.5 ppm corresponding to the 9 protons of NIPAAm (1, 2, and 7) plus the 2 protons of –CH2 (1′). Equation (2) represents the area of methine protons (4 and 5) at 7.06 ppm of VI (A2).

### 2.4. Potentiometric Titration

The pK_a_ values of poly(NIPAAm-co-VI) were determined by potentiometric titration with the 0.1 M HCl solution in the absence and in the presence of 0.1 M NaCl. The experiments were performed at 20 °C with an all-purpose Metrohm 716 DMS Titrino apparatus equipped with a dosing unit and a combined glass electrode. Previously, the electrode was standardized at 20 °C. 

### 2.5. Determination of the Lower Critical Solution Temperature

The lower critical solution temperature (LCST) was evaluated by continuously monitoring the absorbance (at 450 nm) of an aqueous solution of the copolymer at different temperatures. The poly(NIPAAm-co-VI) solutions (1%, *w*/*v*) were obtained by solving the copolymer in distilled water and standard phosphate buffer solution (PB) (pH = 7.4, 6.6, 6.0, 5.5, and 5.0). Every 10 min, the temperature was raised by 0.2 °C. The experiments were completed using an UV-VIS Specord 200 spectrophotometer (Analytic Jena, Jena, Germany) equipped with a temperature controller. The LCST was considered to be the temperature at which the absorbance attains the value of 1.0. 

### 2.6. Synthesis of Poly(NIPAAm-co-VI) Microspheres and Nanospheres

The preparation of poly(NIPAAm-co-VI) microspheres was performed by the w/o solvent evaporation method. For example, 0.2 g of copolymer and 0.005 g of DXR were first dissolved in 0.05 M HCl (4 mL); then, the solution was dripped in cyclohexane (60 mL) containing soybean lecithin (0.2 g) as the dispersing agent. The mixture was homogenized for 5 min at 12,000 rpm using an Ultra-Turrax T 25 homogenizer (IKA Labortechnik, Staufen, Germany). Subsequently, the water was evaporated from the droplets under magnetic stirring at 800 rpm for 6 h at 30 °C and for 60 min at 45 °C. Periodically, 5 mL of fresh cyclohexane were added to keep the volume of the continuous phase constant. Finally, the microspheres were washed with diethyl ether (3 × 50 mL) and separated by ultracentrifugation at 10,000 rpm. Poly(NIPAAm-co-VI) nanospheres loaded with DXR (DXR-NS) were obtained in situ by disintegration of already formed microspheres after incubation in PBS (pH = 7.4) at a temperature situated above the LCST (36 °C) of the linear copolymer.

### 2.7. Morphological Analysis

The dimension and the morphology of poly(NIPAAm-co-VI) microspheres were assessed by environmental scanning electron microscopy (ESEM, type Quanta 200, FEI Company, Brno, Czech Republic). The size as well as the size distribution of microspheres were determined by inspecting ESEM photomicrographs, taking into account at least 200 microspheres. Microspheres were measured and counted and each fraction was compared with the total number of microspheres.

### 2.8. Hydrodynamic Size in Physiological Buffer 

For biological investigations, the DXR-loaded microspheres were suspended in phosphate buffer saline (PBS), pH: 7.4, to obtain a stock solution of 1 mg/mL, and kept at a temperature above 36 °C to prevent particle destabilization. The particle suspension was diluted 1:1000 in PBS and investigated for size by dynamic light scattering (DLS) method using size distribution by intensity, with a Zetasizer Nano ZS (ZEN 3600) (Malvern Instruments, Malvern, UK). The size Standard Operating Procedure (SOP) measured the scattered light intensities at an angle of 173°, using the following physico-chemical parameters: viscosity of dispersant (PBS): 0.8882 cP; refractive indices for PBS: 1.335 and for material: 1.59. One individual record represents the average of 22 measurements per sample. The reported data comprise the Z-averages of intensity distributions (nm) and the PDI (polydispersity index) of at least three individual recordings.

### 2.9. Transmission Electron Microscopy (TEM)

TEM images were obtained using a Hitachi High-Tech HT7700 microscope (Hitachi High-Technologies Corporation, Tokyo, Japan) operating in “high contrast” mode and at a 100 kV acceleration potential. Samples were prepared by dripping a suspension of MS-DXR in PBS (pH = 7.4) at 36 °C to 300 mesh copper grids, coated with carbon, previously heated at 36 °C. The grid was kept in an oven at the same temperature until the solvent was completely evaporated. 

### 2.10. Loading/Release Studies

The percentage of DXR in microspheres was determined by dissolving 10 mg of DXR-loaded microspheres in distilled water (5 mL) below the LCST (20 °C). The solution of copolymer and drug was introduced in a dialysis bag and the bag was then immersed in 45 mL water. The concentration of the drug in the supernatant was determined by measuring the absorbance with an UV-Vis spectrophotometer using a calibration curve. The loading capacity was determined according to Equation (3):(3)$$Loading(\%)=\frac{W_{\mathrm{DXR}}}{W_{\mathrm{microsph}}}\times 100\;$$
where *W*_DXR_ and *W*_microsph_ represent the weight of DXR and of loaded microspheres, respectively.

The relative entrapment efficiency of DXR was calculated as the ratio between the actual and the theoretical DXR content according to Equation (4):(4)$$Efficiency(\%)=\frac{{\mathrm{DXR}}_{a}}{{\mathrm{DXR}}_{t}}\times 100\;$$
where *DXR*_a_ is the actual amount of DXR in microspheres and *DXR*_t_ is the theoretical amount of DXR in microspheres.

In vitro release studies were performed at 36 °C in phosphate buffer with different pH (7.4, 6.6, 6.0, 5.5, and 5.0). In a dialysis bag, 100 mg of DXR-loaded microspheres were dispersed in 10 mL of buffer, and the bag was immersed in another 40 mL of the same buffer. At well-established intervals, samples were taken and the absorbance of the solutions was measured (λ = 482 nm) using a UV-VIS spectrophotometer. After each sampling, the same amount of fresh buffer was added to maintain constant the volume of release fluid. 

### 2.11. Cell Culture

The human hepatocarcinoma (HepG2) cell line was grown in Dulbecco’s modified Eagle’s medium (DMEM) containing 0.45% glucose, while the human adenocarcinoma alveolar basal epithelial (A549) cell line was cultured in DMEM medium containing 0.1% glucose. Both cell lines were purchased from American Type Culture Collection (ATTC), Manassas, VA, USA. All media were supplemented with 10% fetal bovine serum (FBS) and 1% antibiotics (100 U/mL penicillin and 100 μg/mL streptomycin) and maintained in cell culture plates in an incubator at 37 °C and 5% CO_2_.

### 2.12. In Vitro Cytotoxicity Assay

#### 2.12.1. Colorimetric Assay

The effect of different concentrations of free DXR, NS-DXR, or plain NS on cell viability was assessed using the XTT reagent [29]. The assay is based on the production of water-soluble formazan after cleavage of the XTT tetrazolium salt in viable cells. HepG2 and A549 cells were seeded for 48 h at a density of 10^4^ cells/well on flat-bottom 96-well plates and incubated with various concentrations of NS or NS-DXR (3.9 ÷ 125 μg/mL) and free DXR, at the corresponding concentrations as entrapped into MS-DXR (0.078 ÷ 2.5 μg/mL) for 24 h. At the end of the incubation interval, each well was incubated for 2 h at 37 °C with the XTT reagent and phenazine methosulfate (PMS) mixture. Then, the absorbance at 450 nm was measured using the Tecan Infinite M200Pro microplate reader. Cell viability was expressed as % of untreated cells (control) considered 100% viable. The IC50 (half maximal inhibitory concentration, namely the concentration of a cytotoxic compound that induces 50% inhibition of the cell proliferation) was determined by plotting the percentage of cell growth inhibition versus the compound concentration.

#### 2.12.2. Bioluminescent Assay

The cytotoxicity induced by the exposure of HepG2 and A549 cells to different concentrations of free DXR, NS-DXR, or plain NS (identical concentration of nanospheres as NS-DXR, but without DXR) was determined using the ToxiLight assay kit (Lonza Bioscience, Basel, Switzerland). The assay uses luminescent detection to measure the levels of adenylate kinase (AK) released by the damaged cells into the culture media. 

HepG2 and A459 cells were seeded in a 96-well culture plates at a density of 10^4^ cells/well and after 48 h the cells were treated with various concentrations of NS or NS-DXR (3.9 ÷ 125 μg/mL) and free DXR, at the corresponding concentrations as entrapped into NS-DXR (0.078 ÷ 2.5 μg/mL) for 24 h. Then, the culture medium was collected, and the cytotoxicity was assessed using the bioluminescent kit according to the manufacturer’s instructions. The results were expressed as fold change relative to the results obtained for the untreated control cells.

### 2.13. In Vitro Cellular Uptake of NS-DXR by Tumor Cells 

Based on the fluorescent properties of DXR [30], we investigated the internalization of nanospheres loaded with DXR by two cancer cell lines (HepG2 and A549 cells) using fluorescence microscopy. Cellular uptake of NS-DXR compared to free DXR was determined by treating HepG2 and A459 cells with two concentrations of DXR (1.25 and 2.5 μg/mL), either free or loaded into NS-DXR. The nanoparticle concentrations corresponding to 1.25 and 2.5 μg/mL of loaded DXR are 62.5 and 125 μg/mL, respectively. For both cell lines, 10^4^ cells were grown for 48 h on a 96-well culture plate. Then, the cells were incubated for 6 and 24 h at 37 °C with NS-DXR or free DXR in a complete medium. Cells were then washed with phosphate-buffered saline (PBS), fixed with 4% paraformaldehyde (PFA), stained with 100 ng/mL phalloidin-FITC for actin cytoskeleton and 2 µg/mL 4′,6-diamidino-2-phenylindole (DAPI) for cell nuclei. The intracellular localization of DXR after incubation of the cells to NS-DXR or free DXR was examined with the Inverted Microscope Olympus IX81 (40× magnification objective), and the representative images were taken using the following filter cubes: U-MNG2 filter, λ excitation/emission: 555/580 nm to detect DXR-derived red fluorescence; U-MNU2 filter, λ excitation/emission: 345/478 nm to detect the DAPI-derived blue fluorescence; and U-MNB2 filter, λ excitation/emission: 494/518 nm to detect FITC-derived green fluorescence.

To quantify the DXR in HepG2 and A549 cells, the intensity of red pixels, corresponding to DXR fluorescence, was normalized to the number of nuclei visualized by DAPI staining (blue) using images acquired with the 10× objective of the microscope. The images were processed using ImageJ software version 1.8.0 (National Institutes of Health (NIH), Bethesda, MD, USA). The data were expressed as the mean ± standard deviation (SD) of three independent experiments performed in triplicate. Thus, for each experimental condition 3 wells/experiment and 3 fields/well were used.

### 2.14. Animals

Sixteen-week-old male C57BL/6J mice (Stock No: 000664) from The Jackson Laboratory (Bar Habor, ME, USA) were used to investigate the hemocompatibility and the biodistribution of NS-DXR compared to free DXR. Mice were kept in a specific pathogen-free facility at 24 °C in a temperature-controlled chamber with a 12 h light/dark cycle and had access to a standard rodent diet and water ad libitum. The experiments were approved by the Ethics Committee of the Institute of Cellular Biology and Pathology “Nicolae Simionescu” and by the National Sanitary Veterinary and Food Safety Authority authorization no. 497/10.02.2020 conducted following the EU directive 2010/63/EU on the protection of animals used for scientific purposes.

### 2.15. In Vitro Hemocompatibility Test

To evaluate the blood compatibility of NS-DXR compared to plain NS and free DXR, the hemolysis was investigated as previously reported [31,32]. The erythrocytes were isolated from the blood collected in EDTA tubes by cardiac puncture from C57BL6J mice. The blood samples were centrifuged for 15 min, at 1000× *g* and 4 °C and the plasma was carefully separated, pooled, and stored at 20 °C for further analysis. The erythrocyte mass was pooled and incubated at a 1:10 ratio in PBS containing different concentrations of NS and NS-DXR (ranging from 0.0078 to 1 mg/mL) or the corresponding free DXR (0.156–20 μg/mL). After incubation for 1 h, on an orbital shaker at 37 °C (OHAUS, Parsippany, NJ, USA), the samples were centrifuged to sediment the intact erythrocytes, and the hemoglobin released in the supernatants was measured at 540 nm with a TECAN Infinite M200Pro plate reader. As the negative control, the erythrocytes incubated with PBS were used, while the positive control (100% hemolysis) was obtained by the erythrocyte’s incubation with 0.5% Triton X-100 (TX-100).

### 2.16. In Vivo Biodistribution and Hemocompatibility Studies

The C57BL6 mice received a retro-orbital injection of NS-DXR (150 mg NS/3 mg DXR/kg body weight, n = 4 mice) and free DXR (3 mg DXR/kg body weight, n = 3 mice) or PBS (as control for fluorescence background, n = 2 mice). At 1 h after administration, the mice were anesthetized with ketamine/xylazine and the blood was collected by cardiac puncture. To remove blood, perfusion through the left ventricle with PBS containing Ca^2+^ and Mg^2+^ was performed. The urine samples were collected from the mouse bladder. The brain, lungs, heart, liver, spleen, and kidneys were harvested and the DXR fluorescence was examined by ex vivo imaging, using the IVIS Spectrum Caliper 200 system with excitation/emission filter pairs: 500 nm/600 nm. For the spectral unmixing option, to discriminate between tissue autofluorescence and DXR fluorescence, the following excitation/emission filter pairs: 500 nm/580 nm, 500 nm/600 nm, 500 nm/620 nm, and 500 nm/640 nm were used. The fluorescent radiant efficiency [fluorescence emission radiance per incident excitation intensity (p/sec/cm^2^/sr)/(μW/cm^2^)] of DXR was quantified using the region of interest (ROI) option of Living Image 4.3.1 software and the resulting intensities were reported to the corresponding organ weight (g). 

Additionally, the DXR concentration was measured in plasma and urine samples, using a validated fluorescent quantitative method adapted from A. Motamarry et al. [33]. Briefly, 50 μL of undiluted plasma sample or 50 µL of urine (diluted in PBS at a 1:4 ratio) were added in a black 96-well plate. The fluorescence intensity of DXR in the plasma and urine samples was measured using the TECAN Infinite M200Pro, at λex = 480 nm and λem = 590 nm. A standard curve of known DXR concentrations (0.625 ÷ 20 µg/mL) was used to determine the concentration of DXR in plasma and urine. In addition, the erythrocytes isolated, as described in Section 2.13, from mice treated with NS-DXR and DXR were investigated for aggregation by light microscopy. Images were taken using the 40× objective of the Olympus IX81 Microscope. 

### 2.17. Statistical Analysis

Data were expressed as mean ± standard deviation (SD). Statistical significance was determined by GraphPad Prism 8 software (GraphPad Software, Version 8, San Diego, CA, USA) using the unpaired two-tailed *t*-test and *p* < 0.05 was considered statistically significant. 

## 3. Results and Discussion

### 3.1. Preparation and Characterization of Poly(NIPAAm-co-VI) Copolymer

As it is well-known, poly(NIPAAm) is one of the most used polymers in biomedical applications because it possesses thermosensitive characteristics around the human body temperature [23]. The copolymerization of NIPAAm with pH-sensitive monomers forms copolymers that display both temperature and pH sensitivity. Here, copolymers of NIPAAm with the pH-sensitive VI were synthesized by free radical polymerization in 1,4-dioxane using AIBN as initiator. The resulted poly(NIPAAm-co-VI) copolymers were solubilized in distilled water and purified by dialysis to remove the unreacted monomers, low molecular weight copolymers, and water soluble residues; then, the co-monomer composition was determined by ^1^H-NMR analysis (Figure 1A). 

As shown in Table 1, the increase in VI in the initial mixture leads to the increase in VI in the copolymer; however, the proportion of VI in the copolymer is slightly lower than that in the initial mixture (samples VI_1_–VI_3_).

### 3.2. Determination of the pK_a_


VI was chosen to be copolymerized with NIPAAm because it contains amino groups that ionize at a pH situated at the boundary between physiological and pathological conditions. Moreover, the protonation/deprotonation of VI modulates the thermosensitive properties of the poly(NIPAAm-co-VI) and lastly the solubility/insolubility of the copolymer. Since the solubility properties control the release place of the drug, it was absolutely necessary to determine the pK_a_ of copolymer. The pK_a_ value was determined by potentiometric titration of poly(NIPAAm-co-VI) (sample VI_2_) with the 0.1 M HCl solution (Figure 1B); then, the dependence of the p*K*_a_^app^ on the ionization degree was represented (Figure 1C). The potentiometric titration was also performed in the presence of 0.1 M NaCl, simulating the ionic strength of the physiological fluids.

The acid dissociation constant was calculated according to Equation (5): *K*_a_^app^*=* [*VI*] (*a_H_*)/[*VIH*^+^](5)
where [*VI*] and [*VIH*^+^] signify the concentration of deprotonated and protonated VI in aqueous solution, and *a_H_* is the activity of the bulk protons. 

The p*K*_a_^app^ was calculated using the Henderson–Hasselbach Equation (6):p*K*_a_^app^ = pH + log(α/(1 − α))(6)

The dissociation degree, *α*, was calculated according to Equation (7):*α* = *n*_HCl_/*P*_tot_(7)
where *n*_HCl_ indicates the moles of HCl added to the polymer solution and *P*_tot_ are the total moles of imidazole residues in solution. The values of the intrinsic acid dissociation constants (p*K*_a_^0^) were obtained by extrapolation of p*K*_a_^app^
*=* f(*α*) curves at α = 0 (Figure 2B). For poly(NIPAAm-co-VI), the p*K*_a_^0^ was found to be 5.6 in pure water and 6.0 in the presence of added electrolyte (0.1 M NaCl); these values being in accordance with the values found in the literature for poly(N-vinylimidazole) [34]. 

### 3.3. Phase Transition Characterization

As it was previously stated, the copolymerization of NIPAAm with VI had as the main objective to produce copolymers with ionizable groups at the pH located at the frontier between physiological and pathological conditions. The protonation/deprotonation of VI modulates the solubility properties of the copolymer that are further exploits for the transport and release of the drug at the tumor site. As follows, LCSTs of the copolymers under different simulated physiological conditions were determined (Table 1). As a general observation, the LCST increases with the increase in the amount of VI in copolymer, suggesting that VI is a hydrophilic co-monomer regardless of the ionization state. Additionally, for all copolymers the LCST increases with the decreasing of the pH from 7.4 to 6.6, 6.0, 5.5, and 5.0, indicating an increase in the hydrophilicity of the copolymer due to the different degrees of protonation of imidazole moieties. It must be underlined that under simulated physiological conditions (PB at pH 7.4), all copolymers possess LCST values below the human body temperature (37 °C); therefore, they are not soluble in physiological fluids and from this point of view they are excellent candidates for drug transport. However, only some of these copolymers become soluble under the pH conditions of the tumors. For example, sample VI_1_ is not soluble in pH 7.4 but neither in any pH characteristic of tumors indicated in Table 1 (6.6, 6.0, 5.5, and 5.0). On the contrary, sample VI_2_ is not soluble in pH 7.4 but is soluble in pH 5.5 and 5.0 at human body temperature. Moreover, sample VI_3_ is stable in pH = 7.4 but becomes soluble in pH 5.5, 5.0, and even 6.0, covering a wide range of pH values characteristic of different tumor regions.

### 3.4. Synthesis of DXR-Loaded Poly(NIPAAm-co-VI) Microspheres (MS-DXR) and Nanospheres (NS-DXR) 

Microspheres from poly(NIPAAm-co-VI) (sample VI_2_) were synthesized using a completely new approach to the w/o solvent evaporation method. Unexpectedly, the continuous phase is the volatile solvent cyclohexane and the dispersed phase is the less volatile water. The choice of this binary solvent system was made for the following reasons: (i) solvents are immiscible; (ii) water is the best solvent for DXR; and (iii) cyclohexane molecules that evaporate easily due to solvent volatility entrain less volatile water molecules reducing the preparation time considerably. Preliminary tests using the mineral oil as continuous phase demonstrated that acceptable microspheres are obtained only after 5 days of water evaporation (data not shown). By replacing the mineral oil with cyclohexane, the preparation time of the microspheres was substantially reduced to 6 h. In fact, the vaporization of cyclohexane favors the evaporation rate of water, even if cyclohexane must be added periodically to keep the w/o ratio constant. It must be underlined that microspheres were prepared in slightly acidic solution that helps to increase the LCST of copolymer solution and perform evaporation of the water at higher temperature but under LCST. As shown in Figure 2A, poly(NIPAAm-co-VI) particles loaded with DXR display a round shape and a rough surface (Figure 2B); they have a relatively large size distribution with a diameter ranging between 500 nm and 8 µm (Figure 2C). Note that the microspheres were produced from an aqueous solution successively using two completely different emulsification rates: an initial rate of 12,000 rpm, which results in the production of nanometer-sized droplets coated with the surfactant, followed by a rate of 800 rpm which results in coagulation of these nanodroplets into solid microspheres of micrometric dimensions. This procedure results in microspheres made up of aggregated nanospheres delimited by the surfactant that disintegrates in contact with physiological fluids. The disintegration takes place because soybean lecithin is a very hydrophilic surfactant, and in aqueous media, it allows water to enter into microspheres and causes their disintegration.

The resulting nanoparticles (noted hereafter as NS-DXR, the particles formed from nascent MS-DXR after their dispersion in physiological fluids at 36 °C) are characterized by a mean diameter of ~250 nm and a polydispersity index (PDI) of 0.2, indicating a homogeneous population of nanoparticles (Figure 2D), confirmed also by TEM image (Figure 2E). In the same conditions, the nanoparticles possess a zeta potential of −9.57 mV. 

### 3.5. DXR Loading/Release 

As it was previously stated, DXR-loaded microspheres were prepared in slightly acidic solution because both the copolymer and drug have the maximum solubility in this medium. In this way, the diffusion of the drug in the continuous phase is reduced. The percentage of entrapped DXR was found to be 2.0% (*w*/*w*) with an encapsulation efficiency of 82.0%. The high encapsulation efficiency was expected since the polymer is not soluble in cyclohexane and the drug diffusion is limited just at the interface between the two phases. 

Release studies were performed under simulated physiological conditions (PB at pH = 7.4) taking into account that DXR-loaded nanospheres circulate in the bloodstream for a while (Figure 2F). In addition, release tests were conducted at low pH simulating the endosomal (pH = 6.0–6.6 for early endosome; pH = 5.0–6.0 for late endosome) and lysosomal (pH = 5.0) conditions after particle internalization. 

In simulated bloodstream conditions (PB at pH = 7.4), the LCST value of poly(NIPAAm-co-VI) (sample VI_2_ in Table 1) is 33.1 °C. Therefore, at human body temperature (36 °C) that is situated above the LCST, the copolymer is completely insoluble, as well as the corresponding nanospheres, and the amount of released drug is very low. After 2 h, presumed to be the period spent by the nanospheres in the bloodstream, just 5.43% of the DXR was released very likely reflecting the fraction of the drug that is located on the surface of the nanospheres. In PB at pH = 6.6, the LCST value is 34 °C, again below the human body temperature, and the drug release profile is almost similar to that at pH = 7.4.

In contrast, at pH = 6.0, 5.5, and 5.0, the LCST values are 36.2, 37.7, and 39.7 °C, respectively, all critical temperatures being situated above the human body temperature. As follows at body temperature, the copolymer and consequently the nanospheres solubilize and release the drug in a controlled manner. 

Thus, the release results recommend the NS-DXR as appropriate particles for further in vivo studies.

### 3.6. Biological Tests

#### 3.6.1. Cytotoxicity Assessment

In vitro cytotoxicity induced by exposure of HepG2 and A549 cells to various concentrations of NS, NS-DXR, and free DXR for 24 h was measured using XTT (Figure 3A,B) and ToxiLight^TM^ (Figure 3C,D) assays. XTT is a colorimetric method and detects the formazan formed by the cleavage of tetrazolium salt, XTT, in metabolically active cells, and the ToxiLight^TM^ assay is bioluminescent and measures the released adenylate kinase (AK) in the conditioned media, an indicator of cellular damage. For the XTT assay, cell viability was expressed as % of untreated cells (control) considered 100% viable, whereas the data from the ToxiLight^TM^ assay were normalized and presented as a fold change to control cells, exposed to culture medium, considered as 1. 

Both methods have shown that the viability of HepG2 and A549 cells was not affected by the incubation with unloaded NS (Figure 3). This result suggests the cytocompatibility of NS with tested cell lines. Moreover, we found that HepG2 and A549 cells show different sensitivities towards NS-DXR and free DXR. The XTT assay shows a dose–response relation for NS-DXR and DXR in both HepG2 and A549 cells with lower IC50 (half maximal inhibitory concentration) values in the case of HepG2 cells (Figure 3A,B). Thus, IC50 of the NS-DXR on the HepG2 cells was 0.68 ± 0.07 µg/mL DXR, whereas on A549 cells was 1.7 ± 0.8 µg/mL DXR. Free DXR had an IC50 value of 0.58 ± 0.07 µg/mL on HepG2 cells, and 2.0 ± 0.4 µg/mL on A549 cells. The ToxiLight^TM^ assay data (Figure 3C,D) confirm the results obtained by the XTT assay; a reduction in cell viability is mirrored by an increased release of AK from damaged cells. In accordance with XTT results, this method also reveals the higher susceptibility of HepG2 cells to NS-DXR and DXR in comparison with A549 cells. 

Our data showing the sensitivity of HepG2 cells over A549 cells are in line with previous reports on the superior cytotoxicity of the N-(2-hydroxypropyl) methacrylamide (HPMA) copolymer–doxorubicin conjugate in HepG2 cells compared with A549 cells [35]. Additionally, another study showed the susceptibility of HepG2 cells compared to A549 cells when exposed to 90 nm silica nanoparticles, with a lower IC50 value obtained for HepG2 cells [36]. 

The ToxiLight^TM^ assay is a very sensitive method, and it detected a higher AK release rate induced by the incubation of HepG2 cells with NS-DXR compared to free DXR. This observation points out the importance of performing assays based on a different principle to investigate the cytotoxicity of a compound [37]. There might be differences between the cells that are inactive metabolically and dead cells that have lost the plasma membrane integrity. Thus, although the XTT assay showed similar percentages of cell toxicity induced by NS-DXR and free DXR, it appears that the percentage of necrotic cells with damaged membranes is significantly higher when HepG2 cells are exposed to NS-DXR.

#### 3.6.2. In Vitro Cellular Uptake of NS-DXR 

The accumulation of NS-DXR particles and free DXR in HepG2 and A549 cells was investigated using fluorescence microscopy (Figure 4). The fluorescence images revealed that both NS-DXR and free DXR were internalized by the cells in a dose- and time-dependent manner (Figure 4A,C). It can be observed that DXR is mainly localized in the nuclei of HepG2 and A549 cells, consistent with previous studies comprising other DXR-delivery systems [38,39].

The quantification of the NS-DXR or free DXR internalization by the cells was undertaken by calculating the intensity of red pixels originating from DXR fluorescence that was normalized to the number of nuclei (visible by DAPI staining) for each image field (Figure 4B,D). 

In both cell types, NS-DXR and free DXR were extensively taken up at 6 and 24 h of incubation (Figure 4B,D). Still, the fluorescence intensity emitted by DXR, at 6 and 24 h, was higher in HepG2 cells compared to A549 cells for both NS-DXR (more than 200% for HepG2 and 80% for A549 cells vs. controls, *p* < 0.001) and free DXR (more than 150% for HepG2 and 80% for A549 cells vs. controls, *p* < 0.001) (Figure 4B,D). This result may explain the higher cytotoxicity induced by NS-DXR exposure in HepG2 cells compared to A549 cells. 

Significant increases were observed in the internalization of NS-DXR and free DXR by HepG2 cells at 24 h compared to 6 h incubation time for both tested concentrations (*p* < 0.01). Additionally, an approximately 30% increase in DXR uptake was observed for HepG2 cells treated with the lowest concentration of NS-DXR (corresponding to 1.25 µg/mL DXR) compared to free DXR at both 6 and 24 h (*p* < 0.05) (Figure 4B). In the case of A549 cells, quantified fluorescence for DXR either loaded into NS-DXR or free was significantly increased after 24 h incubation compared to 6 h at both concentrations tested (*p* < 0.001) (Figure 4D). Yet, no significant differences were observed between the internalization of DXR from NS-DXR compared to free DXR by A549 cells at both incubation times (Figure 4D). This finding is consistent with the cytotoxicity data and explains the higher sensitivity of HepG2 cells to DXR compared to A549 cells (Figure 3A,B). 

#### 3.6.3. In Vitro Hemocompatibility

As a preliminary step toward the further preclinical investigation of the therapeutic potential of NS-DXR in animal models after intravenous administration, we assessed the hemolytic behavior of NS, NS-DXR, and free DXR. We evaluated the effect of different formulations on erythrocytes membrane integrity by quantifying the hemoglobin released from erythrocytes, isolated from C57BL6 mice, after incubation with NS, NS-DXR, and free DXR (Figure 5A). The results showed that the percentage of hemolysis did not exceed 3.5%, pointing out that the erythrocytes were not affected by the incubation with NS, NS-DXR, and free DXR (Figure 5B). A threshold of 5% hemolysis is considered acceptable for biomaterials, according to International Organization for Standardization (ISO) 10993-4:2017.

#### 3.6.4. In Vivo Biodistribution and Hemocompatibility of NS-DXR 

The localization of NS-DXR and free DXR in different organs, after retro-orbital injection in C57BL6 mice, was investigated by fluorescent optical imaging, based on DXR’s fluorescent properties. Thus, one hour after NS-DXR, free DXR, or PBS administration, the blood was drawn by cardiac puncture and, after thoroughly washing of the vasculature with PBS through the left ventricle, the organs (brain, lungs, heart, liver, spleen, kidneys) were harvested. 

We investigated whether the biodistribution of DXR loaded into NS is different from that of free DXR. The organs were visualized ex vivo by IVIS Caliper 200 imaging system. The fluorescent radiant efficiency measurements using the region-of-interest (ROI) function of Living Image 4.3.1 software were used to quantify the biodistribution of NS-DXR and free DXR. The autofluorescence of organs isolated from mice injected with PBS was subtracted from the ROI values measured for each organ. The spectral unmixing was performed to demarcate the tissue autofluorescence from the specific fluorescence of DXR. The fluorescence emission data show, at 1 h after administration, an accumulation of DXR, either free or encapsulated into NS, in the liver and kidneys (Figure 6A,B). The spectral unmixing evaluation confirms that the measured fluorescence is the specific signal coming from DXR (Figure 6C). A significant increase in the localization of DXR in the kidneys was obtained in the case of free DXR administration as compared with NS-DXR (*p* < 0.05). This result suggests a slower renal clearance of DXR in the case of its formulation into NS, due to the modified pharmacokinetic profile. Moreover, this may signify that a large part of NS-DXR is stable in the blood circulation and strengthen the stability data obtained in in vitro release experiments (Figure 2D). This is in line with the measurements of DXR concentration by measuring the fluorescent signal coming from DXR in plasma and urine samples that showed an increased concentration of DXR in plasma (by ~35%, *p* < 0.05) and a decrease in its concentration in urine (by ~80%, *p* < 0.001) in the case of NS-DXR administration as compared with free DXR (Figure 6D). The decreased fluorescence signal detected by IVIS investigations in the kidneys isolated from mice receiving NS-DXR confirms a significant reduction in renal excretion of DXR in comparison with the excretion of free DXR. 

The administration of NS-DXR and free DXR caused no erythrocyte aggregation, the appearance of erythrocytes showing similarity with the negative control (administration of PBS) (Figure 6E).

A schematic representation of the potential mode of operation of the intelligent drug delivery system after interaction with tumor cells is presented in Figure 7.

In simulated bloodstream conditions, the synthesized microspheres (Figure 7A) disintegrate in monodisperse stable nanoparticles (Figure 7B). After interaction with a tumor cell, the nanoparticles are uptaken through different types of endocytosis (Figure 7C). Then, the particle internalization continues by their passage from endosomes to lysosomes where the nanoparticles self-disintegrate in acidic pH and release the payload. 

## 4. Conclusions

The bio-responsive poly(N-isopropylacrylamide-co-vinylimidazole) copolymer was synthesized and characterized. Microspheres obtained from this copolymer, encapsulating DXR (MS-DXR), were produced. The in vitro and in vivo data showed that: (i) in PBS (pH = 7.4) at 36 °C, MS-DXR disintegrate in small nanoparticles, NS-DXR (average diameter ~250 nm) with a narrow size distribution (PDI = 0.207); (ii) NS-DXR are stable in the conditions similar to that found in the bloodstream (pH = 7.4, T = 36 °C) protecting the drug, but solubilize after internalization (pH = 6.0–5.0) releasing the payload; (iii) the plain particles are cytocompatible and deliver the loaded DXR in a dose- and time-dependent manner in both HepG2 and A549 cells; (iv) the DXR delivered by NS-DXR is mainly localized in the nuclei of HepG2 and A549 cells, similar to free DXR; (v) NS-DXR are internalized at a greater extent by HepG2 cells compared to A549 cells, and this correlates with increased cytotoxicity induced by NS-DXR in HepG2 cells as compared with A549 cells; (vi) NS-DXR particles do not cause hemolysis and erythrocytes aggregation; and (vii) NS-DXR and free DXR localize in the liver and kidneys of mice, and the loading of DXR into NS resulted in the reduced renal clearance of DXR.

The data suggest that the newly developed poly(NIPAAm-co-VI)-based nanoparticles are biocompatible and may be introduced as carriers for the antitumoral drug, doxorubicin. The NS-DXR might have the potential to deliver DXR to hepatic tumors and, besides, the particulate system can be adapted and endowed with targeting properties by conjugating specific ligands to direct them more efficiently to a certain tumor.

## Figures and Tables

**Figure 1 pharmaceutics-14-00865-f001:**
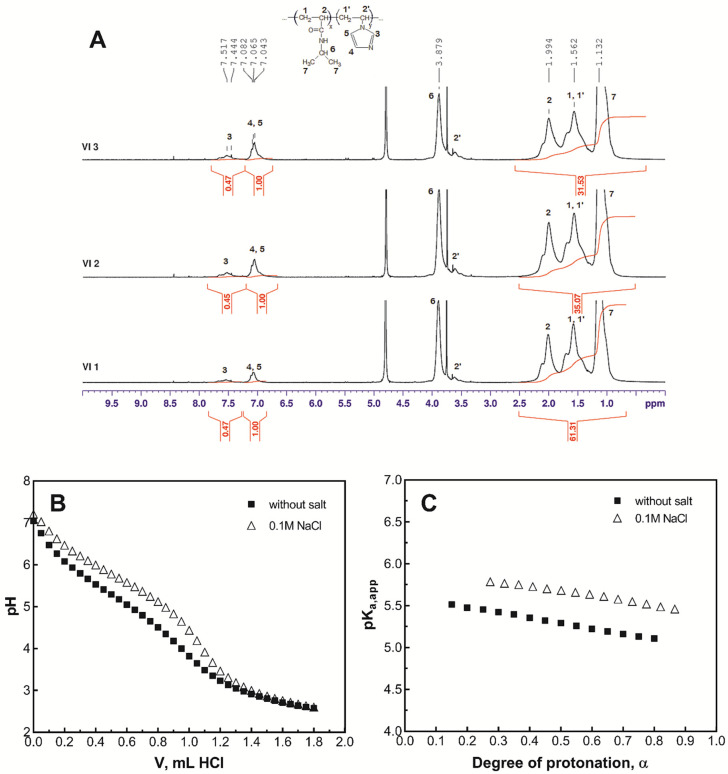
Characterization of poly(NIPAAm-co-VI) copolymers: ^1^H-NMR spectra in D_2_O (**A**); potentiometric titration curves of polymer aqueous solution (sample VI_2_ in Table 1) in the absence (empty symbols) and in the presence (filled symbols) of 0.1 M NaCl (**B**); dependence of p*K*_a,_^app^ on the dissociation degree *α* (**C**).

**Figure 2 pharmaceutics-14-00865-f002:**
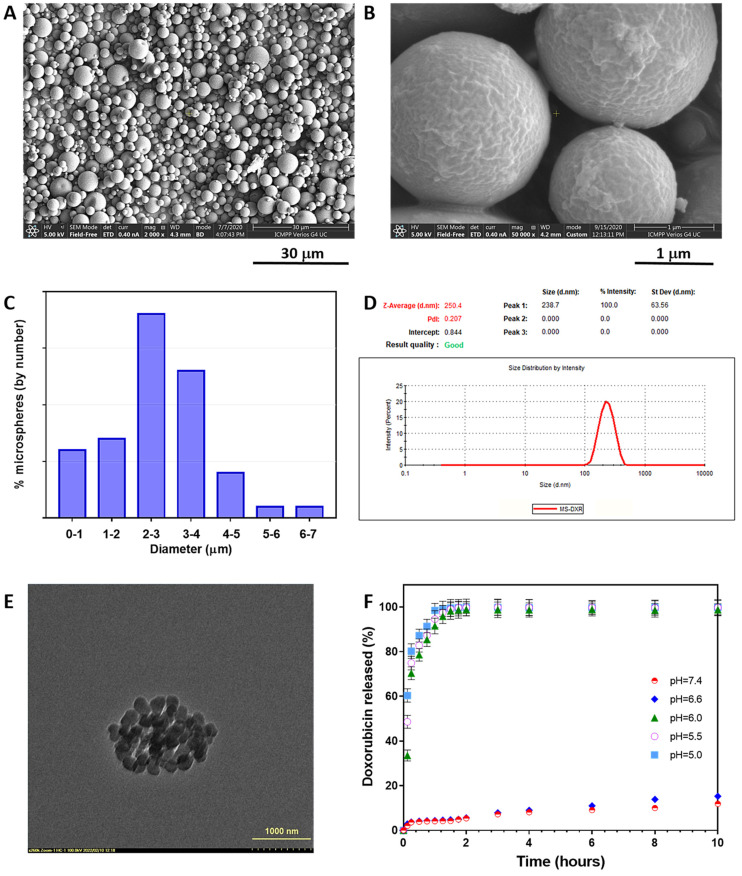
Particle characterization: scanning electron micrographs of DXR-loaded microspheres obtained by w/o solvent evaporation method (**A**,**B**); size distribution of microspheres (**C**); size distribution of NS-DXR in PBS (pH = 7.4) at 36 °C (**D**). TEM image of NS-DXR (**E**). In vitro cumulative release (%) of DXR from NS-DXR under physiological conditions (PB) at pH = 7.4 (circles) and under slightly acidic conditions at pH = 6.6 (diamonds), 6.0 (triangles), 5.5 (empty circles), and 5.0 (squares) (**F**). Where statistical error bars are not shown, they are smaller than the symbols. Results are expressed as means ± standard deviation (S.D.).

**Figure 3 pharmaceutics-14-00865-f003:**
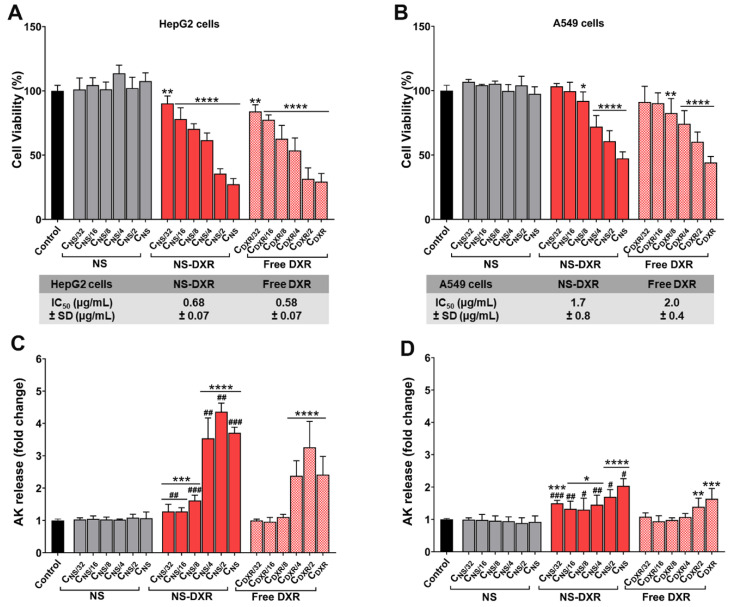
The viability of HepG2 (**A**) and A549 (**B**) cells after exposure to plain NS, NS-DXR, and free DXR determined by XTT assay. The cytotoxicity of plain NS, NS-DXR, and free DXR on HepG2 (**C**) and A549 (**D**) cells determined by ToxiLight^TM^ assay. Cells were exposed for 24 h to various concentrations of NS and NS-DXR (3.9 ÷125 μg/mL, C_NS_ = 125 μg/mL) and the corresponding concentrations of free DXR (0.078 ÷ 2.5 µg/mL, C_DXR_ = 2.5 μg/mL). Results are expressed as means ± standard deviation (S.D.) of three experiments performed at least in triplicates. * *p* < 0.05, ** *p* < 0.01, *** *p* < 0.00, **** *p* < 0.0001 vs. control; # *p* < 0.05, ## *p* < 0.01, ### *p* < 0.001 vs. free DXR.

**Figure 4 pharmaceutics-14-00865-f004:**
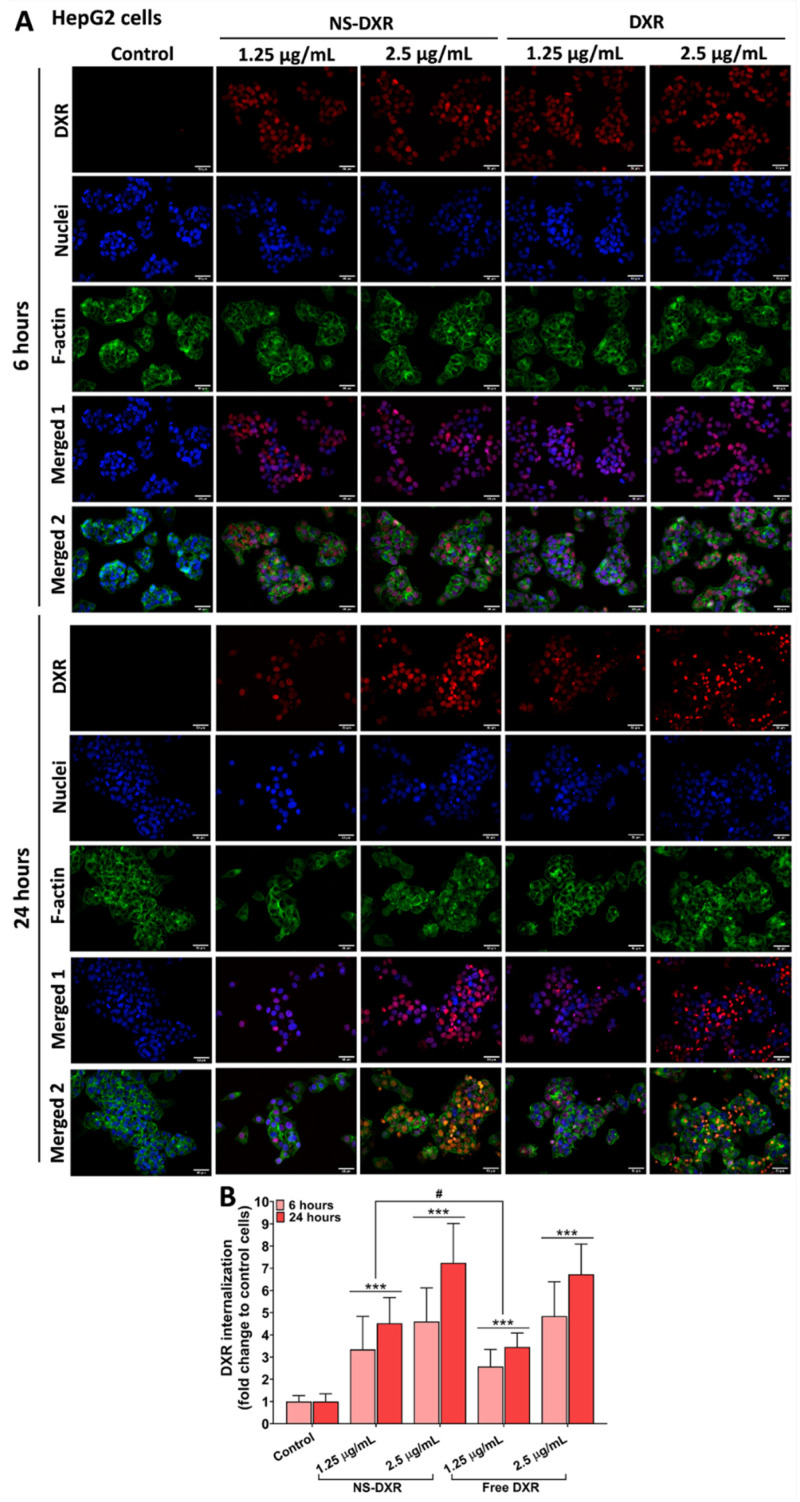
DXR fluorescence visualized in HepG2 (**A**) and A549 cells (**C**) incubated with NS-DXR and free DXR for 6 and 24 h. Cells were exposed to two doses of DXR either free or loaded into NS-DXR (1.25 and 2.5 µg/mL). The nanoparticle concentrations corresponding to 1.25 and 2.5 μg/mL of loaded DXR are 62.5 and 125 μg/mL, respectively. Representative fluorescence microscopy images show the internalized DXR (red), nuclei stained by DAPI (blue) and F-actin, obtained by cell staining with FITC-labeled phalloidin (green). Merged-1 are the merged images of DXR and nuclei, whereas Merged-2 are the merged images of DXR, nuclei, and F-actin. Scale bar: 50 μm. Quantification of NS-DXR and DXR uptake by HepG2 cells (**B**) and A549 cells (**D**) expressed as the ratio of red fluorescence to nuclei number. Each point represents a mean of 27 fields. The bar graph shows results expressed as means ± standard deviation (S.D.) of three experiments performed in triplicate. Statistical significance: *** *p* < 0.001, # *p* < 0.05.

**Figure 5 pharmaceutics-14-00865-f005:**
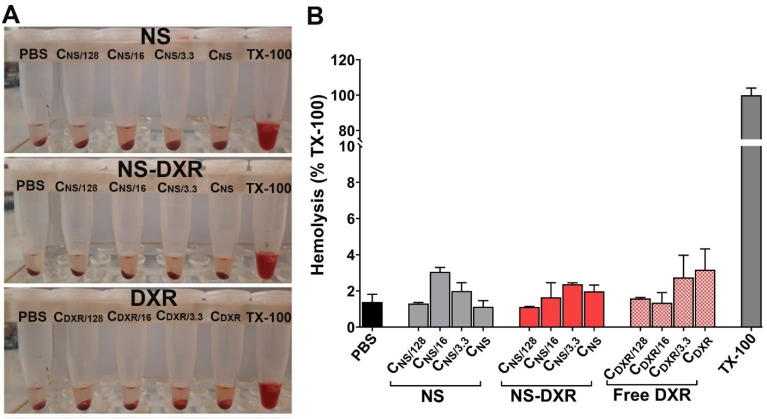
In vitro hemocompatibility evaluation of NS, NS-DXR, and DXR. (**A**) Photographs of the sedimented erythrocytes after their exposure to various concentrations of NS and NS-DXR (0.0078 ÷ 1 mg/mL, C_NS_ = 1 mg/mL), and to the corresponding free DXR concentrations (0.156 ÷ 20 μg/mL, C_DXR_ = 20 µg/mL). (**B**) Quantification of hemolysis by measuring the absorbance of hemoglobin at 540 nm. Results are expressed as means ± standard deviation (S.D.) of one experiment performed in duplicate.

**Figure 6 pharmaceutics-14-00865-f006:**
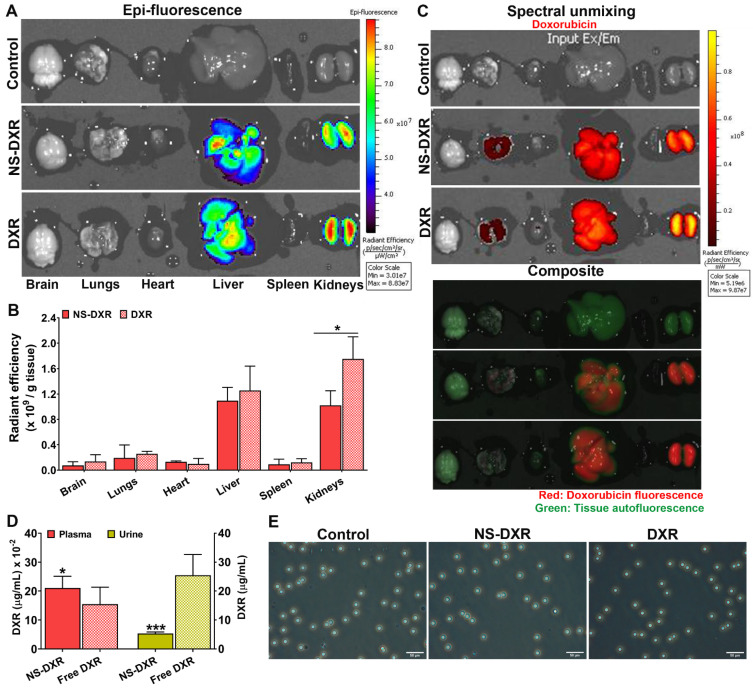
(**A**) Localization of NS-DXR and free DXR in organs harvested from C57BL/6 mice. The analysis was completed 1 h after the retro-orbital injection of NS-DXR (n = 4), free DXR (n = 3), or PBS (n = 2) in mice using the imaging system IVIS Caliper 200, by detection of the DXR fluorescence using the filter set λ_ex_/λ_em_: 500 nm/600 nm. (**B**) Quantification of total radiant efficiency in organs using the region-of-interest option of Living Image software; the resulting intensities were reported to the corresponding organ weight (g). (**C**) Spectral unmixing analysis to delineate the signal specific for DXR fluorescence and tissue autofluorescence. The composite image displays DXR fluorescence (red) and tissue autofluorescence (green). (**D**) Quantification of DXR in plasma and urine of mice, 1 h after administration of 3 mg/kg of free DXR or incorporated into NS-DXR, at λ_ex_ = 480 nm and λ_em_ = 590 nm. * *p* < 0.05, *** *p* < 0.001. (**E**) Evaluation of in vivo hemocompatibility by following the erythrocyte aggregation after the administration of NS-DXR and free DXR. Samples from mice who received PBS were considered the negative control. Scale bar: 50 μm.

**Figure 7 pharmaceutics-14-00865-f007:**
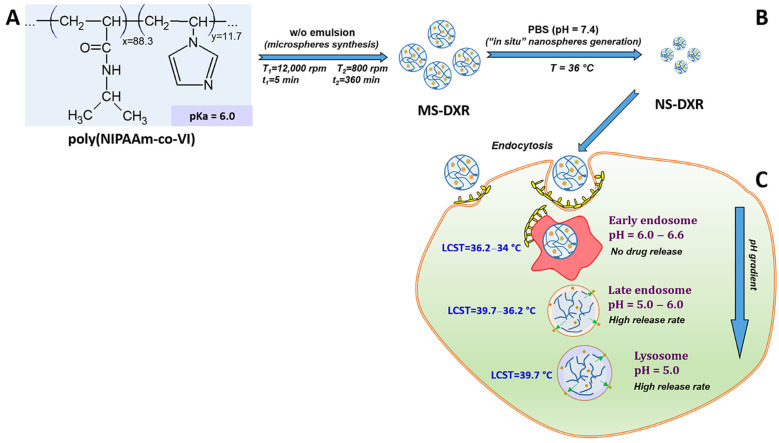
Schematic representation of the potential behavior of DXR-loaded microspheres (MS-DXR) made of poly(N-isopropylacrylamide-co-vinylimidazole) copolymer after suspension in physiological buffers and interaction with a tumor cell. The synthesized MS-DXR (**A**) disintegrate in monodisperse stable nanoparticles (NS-DXR) in conditions simulating the bloodstream (PBS, pH = 7.4, T = 36 °C) (**B**), that are internalized by the cancer cells by endocytosis (**C**). After internalization, the NS-DXR arrives in the late endosomes (pH: 6.0–5.0), where they start to disintegrate and release the drug cargo.

**Table 1 pharmaceutics-14-00865-t001:** Composition of co-monomers in the feed and in the copolymer and dependence of LCST on the co-monomer molar ratio (concentration of copolymer solution was 1%, *w*/*v*).

Sample Code	Co-Monomer Composition					LCST (°C)				
	In the Feed × 10^−3^ M (% mol Ratio)		In Copolymer (% mol Ratio)		H_2_O	pH = 7.4	pH = 6.6	pH = 6.0	pH = 5.5	pH = 5.0
	NIPAAm	VI	NIPAAm	VI						
VI_0_	10	0								
	(100)	(0)	100	0	32.4 ± 0.3	28.7 ± 0.2	30.9 ± 0.2	30.5 ± 0.2	29.3 ± 0.3	31.5 ± 0.3
VI_1_	10	1								
	(90.9)	(9.1)	93.1	6.9	35.0 ± 0.2	32.1 ± 0.2	32.8 ± 0.3	33.4 ± 0.2	34.5 ± 0.3	36.0 ± 0.2
VI_2_ ^a^	10	1.5								
	(86.96)	(13.04)	88.33	11.68	37.0 ± 0.3	33.1 ± 0.4	34.0 ± 0.2	36.2 ± 0.2	37.7 ± 0.3	39.7 ± 0.3
VI_3_	10	2								
	(83.33)	(16.67)	87.15	12.85	38.2 ± 0.2	34.3 ± 0.4	35.0 ± 0.3	37.1 ± 0.3	38.9 ± 0.4	40.6 ± 0.4

Data are the results of two independent experiments; ^a^ M_n_ = 12,560 g/mol, M_w_ = 20,400 g/mol, IP = 1.624 (IP = index of polydispersity).

## Data Availability

All data supporting the findings of this study are available from the corresponding authors upon request.

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
