# Peer review of "Bio-Responsive Carriers for Controlled Delivery of Doxorubicin to Cancer Cells"

_pharmaceutics, 2022, doi:10.3390/pharmaceutics14040865_

Round 1

Reviewer 1 Report

In present form the article is acceptable for publication.

Author Response

Thank you for your appreciation.

Reviewer 2 Report

The article “Bio-responsive micro-carriers for controlled delivery of doxorubicin to cancer cells” is devoted to development of doxorubicin carriers for liver cancer. There are some issues that need to be addressed:

  • Figure 2, A,B: please, provide scale bar and mark with font size similar to article body to make them readable.
  • Figure 4: please, check if inscriptions on figure (NS-DXR, Free DXR) correlate with bars.
  • In the study, the viability of HepG2 and A549 cells after exposure to plain NS, NS-DXR, and free DXR was investigated. The difference in cell viability was noticed. As the developed carriers are sensitive to pH, I wonder if the difference in cell responses is connected with different pH in cell culture medium and in cells? Have the pH levels been monitored in controls and in cultured cells with samples?
  • The scheme in Fig. 7 is just a speculation based on previously published data. The article will be stronger if the authors would summarize their own results.
  • The particle/molecule biodistribution strongly depends on size. Can an increased accumulation of NS-DXR in liver be a result of size-sorting excretion mechanism in the body (if compare size of doxorubicin molecule and NS-DXR carrier)?

Author Response

Comments and Suggestions for Authors

The article “Bio-responsive micro-carriers for controlled delivery of doxorubicin to cancer cells” is devoted to development of doxorubicin carriers for liver cancer. There are some issues that need to be addressed:

  • Figure 2, A,B: please, provide scale bar and mark with font size similar to article body to make them readable.

Answer.  We added the scale bars and marked them.

  • Figure 4: please, check if inscriptions on figure (NS-DXR, Free DXR) correlate with bars.

Answer. We checked and we confirm that is OK.

  • In the study, the viability of HepG2 and A549 cells after exposure to plain NS, NS-DXR, and free DXR was investigated. The difference in cell viability was noticed. As the developed carriers are sensitive to pH, I wonder if the difference in cell responses is connected with different pH in cell culture medium and in cells? Have the pH levels been monitored in controls and in cultured cells with samples?

Answer. We understand the reviewer's concern. We have checked the medium’s pH in the absence and the presence of nanoparticles, and there was no significant change, the pH values being ~ 7.4. Therefore, the destabilization of NS-DXR before internalization is unlikely. The difference in the viability of HepG2 and A549 cells may be explained by the distinct sensitivity of HepG2 cells over A549 cells, as previously reported by others (doi: 10.3109/10717544.2014.911991, doi.org/10.1049/mna2.12105).

  • The scheme in Fig. 7 is just a speculation based on previously published data. The article will be stronger if the authors would summarize their own results.

Answer. We completely modified Fig. 7 according to our experiments. Also, we modified the Figure legend and we added new comments in manuscript according with the new Figure (with red).

  • The particle/molecule biodistribution strongly depends on size. Can an increased accumulation of NS-DXR in liver be a result of size-sorting excretion mechanism in the body (if compare size of doxorubicin molecule and NS-DXR carrier)?

Answer. The reviewer is right. Nanoparticles display distinctive pharmacokinetics and biodistribution compared to drugs that have smaller sizes. The renal clearance of free DXR is higher than when formulated into NS-DXR since nanoparticles evade the 5 nm renal filtration cutoff. As a consequence, the time survival in the circulation increases for DXR encapsulated into nanoparticles compared to free DXR, allowing nanoparticles to accumulate in other tissues. 

Reviewer 3 Report

Manuscript titled " Bio-responsive micro-carriers for controlled delivery of doxorubicin to cancer cells" is a very interesting paper on the drug delivery systems in cancer cells. Methods are clear, results are well described and references are of good quality.

Authors should improve manuscript in several parts:

1) improve the introduction with a proper description of hyaluronic acid-based drug delivery systems. authors should explain the role of cd44 targeting in prostate and mammary drug delivery ( cite 10.1007/s10856-013-4895-4 and 10.1002/jcp.25283) 

2) please add information about the PLGA based drug delivery

3) please add more information on the role of CD44 in cancer cell migration and survival.

Manuscript will be accepted after minor revision

Author Response

Manuscript titled "Bio-responsive micro-carriers for controlled delivery of doxorubicin to cancer cells" is a very interesting paper on the drug delivery systems in cancer cells. Methods are clear, results are well described and references are of good quality.

Authors should improve manuscript in several parts:

1) improve the introduction with a proper description of hyaluronic acid-based drug delivery systems. authors should explain the role of cd44 targeting in prostate and mammary drug delivery ( cite 10.1007/s10856-013-4895-4 and 10.1002/jcp.25283).

2) please add information about the PLGA based drug delivery

3) please add more information on the role of CD44 in cancer cell migration and survival.

Answer. As the reviewer asked, we added the suggested references in the introduction section (page 1 with red). We did not enter into details regarding the PLGA-based drug delivery and the role of CD44 in cancer cell migration and survival since these are beyond the scope of our paper focused on the development of pH/temperature-sensitive copolymer-based carriers. We hope the reviewer will understand this.

Manuscript will be accepted after minor revision

Round 2

Reviewer 2 Report

The images in Figure 4 a, b are unreadable. It is suggested to increase the image size (e.g., to divide the images in (a) and (b) sections into two separated figures).

Author Response

The images in Figure 4 a, b are unreadable. It is suggested to increase the image size (e.g., to divide the images in (a) and (b) sections into two separated figures).

Answer: Although we consider that Figure 4 is not unreadable, since the resolution of the images is good enough to see the details when magnifying the Figure (up to 500 %), we considered the reviewer's suggestion and divided Figure 4 into two different pages. Please, look at the revised Figure 4.